# Immobilization of Soybean Lipoxygenase on Nanoporous Rice Husk Silica by Adsorption: Retention of Enzyme Function and Catalytic Potential

**DOI:** 10.3390/molecules26020291

**Published:** 2021-01-08

**Authors:** Putheary Ngin, Kyoungwon Cho, Oksoo Han

**Affiliations:** Department of Molecular Biotechnology and Kumho Life Science Laboratory, College of Agriculture and Life Sciences, Chonnam National University, Gwangju 500-757, Korea; putheary.ngin@gmail.com (P.N.); kw.cho253@gmail.com (K.C.)

**Keywords:** rice husk silica, lipoxygenase, immobilization, adsorption, oxylipin biosynthesis, octadecanoid pathway, jasmonic acid, plant defense, catalytic efficiency, matrix

## Abstract

Soybean lipoxygenase was immobilized on nanoporous rice husk silica particles by adsorption, and enzymatic parameters of the immobilized protein, including the efficiency of substrate binding and catalysis, kinetic and operational stability, and the kinetics of thermal inactivation, were investigated. The maximal adsorption efficiency of soybean lipoxygenase to the silica particles was 50%. The desorption kinetics of soybean lipoxygenase from the silica particles indicate that the silica-immobilized enzyme is more stable in an anionic buffer (sodium phosphate, pH 7.2) than in a cationic buffer (Tris-HCl, pH 7.2). The specific activity of immobilized lipoxygenase was 73% of the specific activity of soluble soybean lipoxygenase at a high concentration of substrate. The catalytic efficiency (k_cat_/K_m_) and the Michaelis–Menten constant (K_m_) of immobilized lipoxygenase were 21% and 49% of k_cat_/K_m_ and K_m_ of soluble soybean lipoxygenase, respectively, at a low concentration of substrate. The immobilized soybean lipoxygenase was relatively stable, as the enzyme specific activity was >90% of the initial activity after four assay cycles. The thermal stability of the immobilized lipoxygenase was higher than the thermal stability of soluble lipoxygenase, demonstrating 70% and 45% of its optimal specific activity, respectively, after incubation for 30 min at 45 °C. These results demonstrate that adsorption on nanoporous rice husk silica is a simple and rapid method for protein immobilization, and that adsorption may be a useful and facile method for the immobilization of many biologically important proteins of interest.

## 1. Introduction

Soybean lipoxygenase (LOX, EC.1.13.11.12) catalyzes the first step in the octadecanoid pathway, which produces diverse oxylipin derivatives including jasmonic acid (JA). The primary substrates of LOX are α-linolenic acid (LnA) and linoleic acid (LA), which are released from the chloroplast membrane by phospholipase C and then converted by LOX into their corresponding hydroperoxyl derivatives [1,2,3,4]. JA is a ubiquitous plant stress hormone whose synthesis is triggered by various biotic and abiotic environmental stimuli. It has been suggested that JA and its analogues protect plants from damaging insects [5] and that they have potential as noncytotoxic chemotherapeutic agents [6,7,8]. Biosynthesis of JA is initiated by 13-LOX, which oxygenates LnA into 13-hydroperoxy octadecatrienoic acid (13-HPOT). Allene oxide synthase (AOS) converts 13-HPOT to an unstable 12,13-epoxy octadecatrienoic acid (12,13-EOT), which is then metabolized to 12-oxo-phytodienoic acid (12-OPDA) by allene oxide cyclase (AOC). Finally, 12-OPDA is reduced to 3-oxo-2-(2′-pentenyl)cyclopentane-1-octanoic acid (OPC 8:0) by 12-oxo-phytodienoate reductase (OPR), which is converted to JA by three cycles of β-oxidation (Scheme 1).

The initial steps of the JA biosynthetic pathway, including the sequential conversion of LnA to 12-OPDA by LOX, AOS, and AOC, occur in the chloroplast [9,10,11], whereas reduction of 12-OPDA by OPR to OPC and the three steps of β-oxidation occur in peroxisomes [12,13,14]. Numerous efforts have been made to manipulate the biosynthesis of oxylipins in vivo in plants [2,15]. However, these efforts have been met with a number of obstacles, including the fact that the enzymes of the JA biosynthesis pathway are located in different cellular organelles [16,17], and the instability and short half-life of JA biosynthetic intermediates, such as allene oxide [18]. An alternative approach is the enzymatic synthesis of commercially viable JA-related compounds in vitro in solution; this approach also has disadvantages, including the high cost of enzyme purification, low recovery of enzymes for repeat production cycles and generally poor long-term operational stability. A potential third approach is to use surface-immobilized components (e.g., solid state enzymatic synthesis), but this requires an efficient and convenient method to immobilize the required enzyme(s) without impairing its catalytic function. Generally available methods for immobilizing enzymes include adsorption, entrapment, cross-linking or other types of covalent binding reactions. Adsorption and entrapment are physical methods that exploit the molecular interactions between enzymes and the selected matrix, whereas cross-linking or covalent binding are chemical methods that depend on the formation of covalent bonds between enzyme and matrix [19].

LOX has attracted great interest for use in industrial-scale production of its reaction products, which include diverse oxylipin derivatives [20] and lipoxygenase-derived products with antibacterial properties [20,21,22]. The results of numerous efforts to immobilize LOX by various methods have been reported [20,23,24,25]. They include the adsorption of LOX on glutenin and gliadin [26], glass and glass wool [27], and the covalent immobilization of LOX on cyanogen bromide (CNBr), glutardialdehyde (GDA), and oxirane acrylic beads. A number of these studies report considerable loss of enzyme stability and function. Here, we explore the use of nanoporous (also applicable to mesoporous) silica prepared from rice husks as the insoluble matrix for in situ immobilization of LOX, as reported previously for other enzymes [28]. Nanoporous rice husk silica (RHS) is produced by treating rice husk with acid, which leads to carbonization–oxidation of the matrix substance [29]. RHS is readily available for use in commercial applications, is safe, inexpensive, and environment-friendly [30], and has been used previously as an immobilization matrix for in vitro biocatalysis [31,32,33,34]. RHS contains nanopores approximately 4–5 nm in diameter, an average particle size of 50 nm, and functional groups accessible to chemical modification [35]. The nanopores of RHS are a critical surface feature, in that they enhance and promote efficient binding between enzyme and lipophilic substrates during oxylipin synthesis.

In our previous work, the enzymes required for multistep biosynthesis of oxylipins in vitro were successfully covalently linked and co-immobilized on an RHS matrix [33]. However, the covalently immobilized LOX showed relatively low enzyme specific activity, and the efficiency of immobilization depended on the type of linker used for covalent immobilization. This result suggests that enzyme immobilization was accompanied by modification of the functional groups on the RHS matrix surface and/or the enzyme, potentially involving chemical reaction(s) with activator and/or spacer molecules. Therefore, it was reasonable to propose and to test whether adsorption of LOX to the RHS matrix would yield a better outcome [36,37], avoiding undesirable modification of the surface functional group(s) [38,39]. Furthermore, immobilization by adsorption is generally simpler and less expensive, and is expected to allow the retention of higher enzyme catalytic activity than covalent immobilization. Here, we report that soybean LOX was reproducibly immobilized on RHS by adsorption and that RHS-adsorbed soybean LOX was stable, retained high levels of catalytic efficiency, and was resistant to thermal inactivation.

## 2. Results and Discussion

### 2.1. Immobilization of LOX on RHS by Adsorption

Mesoporous materials are useful matrices for in vitro biosynthesis, at least in part because of their porous nature, which increases the amount of surface area available to interact with enzymes, leading to higher enzyme content per unit mass than alternative nonporous reaction matrices [40,41,42,43,44]. In our previous study, oxylipin biosynthetic enzymes, including soybean LOX, rice AOS, and rice AOC, were covalently immobilized on an RHS matrix via glutardialdehyde (GDA) or epichlorohydrin/polyethylene glycol 8000 (ECH-PEG) linkers. The results revealed wide variation in immobilization efficiency, from 49.6% to 92.4% and 24.6% to 51.32% for GDA-linked RHS and ECH-PEG-linked RHS, respectively, and that immobilization efficiency was influenced by the molecular size of the immobilized proteins and the hydrophobicity of the solvent in which the protein was dissolved [33].

To extend our previous study, soybean LOX with a relative molecular mass of 102 kDa and pI of 5.65 [45], the first enzyme in the oxylipin biosynthetic pathway, was immobilized on RHS by adsorption, and the efficiency and capacity of binding was evaluated. Bovine serum albumin (BSA), with a relative molecular mass of 66 kDa and pI of 5.4 [46], was included as a typical globular control protein. A fixed amount (20 mg) of RHS was challenged with 0.2–12 mg LOX (see Section 3). Results were evaluated and are presented in Figure 1, which shows the percentage of adsorbed protein and the ratio (*w/w*) of target protein (LOX or BSA) to RHS, as a function of total added protein (Y axis).

The efficiency of LOX immobilization on RHS was calculated from the ratio of adsorbed protein to total loaded protein. As expected, the immobilization efficiency depended on total protein loaded (LOX or BSA), decreasing hyperbolically as total loaded protein increased (Figure 1A, LOX; Figure 1B, BSA). The binding capacity, calculated as the mass ratio of adsorbed protein to RHS, was linearly proportional to the amount of protein loaded, with constant input of RHS (Figure 1). Furthermore, under the conditions used here, the maximal adsorption efficiency was 55% of total protein input and the adsorption capacity was 7.8% per unit mass of RHS for BSA. Similar results were obtained with soybean LOX, which demonstrated a maximal adsorption efficiency of 50% and a maximal adsorption capacity of 26%. This result suggests that the maximal adsorption capacity per unit mass of RHS varies depending on the molecular size of the immobilized protein and associated interactions between RHS and target proteins.

### 2.2. Effect of Immobilization on RHS by Adsorption on Specific Activity and Kinetic Parameters

To investigate the specific activity of soybean LOX immobilized on RHS by adsorption, LOX specific activity was measured under one of three conditions: (i) free in solution in the absence of RHS (free LOX); (ii) free in solution in the presence of RHS with mixing (unadsorbed LOX); or (iii) immobilized in solid state on RHS (RHS-immobilized LOX). In all three cases, specific activity was measured at a high concentration of substrate (2.5 mM LA). Briefly, soybean LOX (197 μg) was mixed with RHS matrix (2 mg), and unadsorbed LOX was separated from RHS-immobilized LOX by centrifugation. The average amount of RHS-immobilized LOX was 68 μg, indicating 35% adsorption efficiency with a relative mass ratio of LOX to RHS of approximately 0.1, as predicted from the results presented above (Figure 1B). A continuous assay method was used for free LOX and unadsorbed (LOX), while a discontinuous assay was used for RHS-immobilized LOX (see Section 3 for details).

As shown in Table 1, under the experimental conditions used here, soybean LOX immobilized on RHS by adsorption retained 73% of the specific activity of its free form. In contrast, soybean LOX covalently immobilized on RHS only retained 2.5% or 23% of the specific activity of free soybean LOX for two different linkers [33]. This represents a remarkable improvement, especially considering that the specific activity of unadsorbed LOX was 83% of free LOX, presumably by partial inactivation during binding with RHS.

These results suggest that immobilization by adsorption may be a practical, effective, and efficient method for immobilizing and preserving the function of soybean LOX, and potentially other enzymes, on inert matrices such as RHS. Additional studies of RHS-adsorbed soybean LOX were carried out using the discontinuous ferrous oxidation–xylenol orange assay method to quantify the kinetic parameters of free and RHS-adsorbed LOX. This method was selected because of the low K_m_ of RHS-adsorbed LOX (see below). The results are shown in Table 2.

Interestingly, the Michaelis–Menten constant (K_m_) of RHS-adsorbed soybean LOX was 49% of the K_m_ of free soybean LOX. This result suggests that RHS could be a favorable matrix for binding the LOX substrate. For example, if the formation of substrate micelles was suppressed and the effective concentration of the substrate increased, adsorption of the substrate to RHS would be predicted to increase [47,48,49,50]. Catalytic turnover (k_cat_) decreased to 11% of the k_cat_ of free soybean LOX, as expected from the decrease in K_m_, indicating the stabilization of the ES (enzyme-substrate) complex. The catalytic efficiency (k_cat_/K_m_) of RHS-adsorbed soybean LOX was 21% of the k_cat_/K_m_ of free soybean LOX. These data suggest that low K_m_ and increased affinity for its substrate could play a role in the large decrease in catalytic turnover observed here for RHS-adsorbed soybean LOX [51]. Furthermore, the results suggest that low catalytic efficiency following immobilization or adsorption of soybean LOX on RHS may be unavoidable at a low concentration of substrate. One caveat worth mentioning is that direct comparison of the kinetic parameters of an immobilized enzyme system with the same enzyme system in solution (i.e., free) is somewhat controversial because the two systems are expected to differ with respect to enzyme mechanism [24,51]. This is particularly true with regard to substrate binding because of the influence of RHS nanopores on protein/matrix and protein/substrate binding interactions [24]. In this regard, it should also be mentioned that the effective concentration of substrate is generally considered to be higher for an immobilized than for a soluble enzyme system, potentially exceeding the saturation point.

### 2.3. Desorption Kinetics of RHS-Adsorbed Soybean LOX

The kinetic stability (i.e., desorption kinetics) of RHS-adsorbed soybean LOX was examined over several weeks in a storage buffer. Because the surface of RHS is rich in anionic functional groups derived from sialic acid with pI value of 2.9 [52], desorption kinetics were analyzed and compared in a cationic buffer of 50 mM Tris-HCl (pH 7.2), and an anionic buffer of 50 mM sodium phosphate (pH 7.2). The amount of desorbed LOX was 70 μg (28%) in Tris-HCl (pH 7.2) and 17 μg (7%) in 50 mM sodium phosphate (pH 7.2), such that 72% or 93% of the enzyme input was retained on RHS in the two buffer systems, respectively (Figure 2). These data indicate that an anionic buffer is preferred to support the binding of soybean LOX to RHS, most likely because anionic features on the RHS surface, such as sialic acid, play a role in binding LOX to this matrix. In contrast, this type of interaction would likely be suppressed in the presence of a cationic buffer. Figure 2 also shows that the desorption kinetics of RHS-adsorbed LOX is hyperbolic with maximal desorption of 56 µg (23%) soybean LOX in a sodium phosphate buffer and maximal desorption of 155 μg LOX (55%) in a Tris-HCl buffer, under the assay conditions used here. Therefore, the stability of RHS-adsorbed soybean LOX is greater in the sodium phosphate buffer than in the Tris-HCl buffer. Furthermore, the desorption kinetics reached saturation after 3 weeks in sodium phosphate (pH 7.2) and after 5 weeks in 50 mM Tris-HCl (pH 7.2). The results in Figure 2 suggest that the anionic buffer works better than the cationic buffer for maintaining the stability of RHS-adsorbed LOX.

### 2.4. pH Dependence of Soybean LOX Adsorption on RHS

Because ionization of the functional groups on the RHS surface, such as sialic acid, may play an important role in the adsorption of soybean LOX, pH was examined as a modulating factor for enzyme immobilization by adsorption on RHS. For this purpose, four buffer systems were tested, covering a wide pH range from 5.06 to 11.4 (see Section 3). All buffers were anionic, except the Tris-HCl buffer which was used at pH 8.03 and 8.57. The pH dependence of the adsorption percentage of soybean LOX on RHS is shown in Figure 3. A maximal binding efficiency of 60% was achieved at pH 7.2 in 50 mM sodium phosphate. The slight increase in adsorption percentage at a high pH could be due to the precipitation of denatured LOX under harsh conditions.

### 2.5. Reusability of RHS-Adsorbed Soybean LOX

The most important advantage of immobilizing an enzyme on a matrix surface is that it allows the enzyme to be used repeatedly, achieving multiple reaction cycles with increased thermal stability [53]. Here, the operational stability of RHS-adsorbed soybean LOX, as indicated by the relative enzyme specific activity, was analyzed over five sequential reaction cycles and compared with that of covalently RHS-immobilized soybean LOX [33]. As expected, the specific activity of RHS-adsorbed soybean LOX gradually decreased with each successive reaction cycle (Figure 4).

High enzyme specific activity (90%) was retained through four cycles, but it dropped to 59% in the fifth cycle. Previously [33], covalently RHS-immobilized soybean LOX was reported to retain 70% to 100% of the initial activity after five cycles, depending on the linker employed for covalent immobilization. The results indicate that covalent immobilization performs better than adsorption with respect to the reusability of soybean LOX. This can be explained by the noncovalent character of adsorption, where a minimum basal rate of adsorption/desorption (i.e., reversible binding) of soybean LOX to RHS is expected to occur. The data in Table 1 are consistent with this explanation, as the activity of unadsorbed soybean LOX is lower than that of free soybean LOX. Nevertheless, immobilization by adsorption remains practical because the specific activity of RHS-adsorbed soybean LOX is higher than that of covalently immobilized soybean LOX [33].

### 2.6. Dependence of Specific Activity of RHS-Immobilized Soybean LOX on Temperature

Immobilized enzymes are generally more resistant to thermal denaturation and inactivation than free enzymes in solution because the thermal mobility of an enzyme is restricted by immobilization [53]. The thermal stability of a solid-state enzyme system is practically important because most globular enzymes in solution are heat-labile. As shown in Figure 5, the optimal temperature for free and immobilized soybean LOX was 35 °C; however, the specific activity of free soybean LOX decreased to 45% at 45 °C, while the specific activity of RHS-adsorbed soybean LOX only decreased to 70% at 45 °C. The observed resistance to thermal inactivation of RHS-adsorbed soybean LOX supports the merit and the practicality of using soybean LOX for enzymatic synthesis of oxylipins.

## 3. Materials and Methods

### 3.1. Materials

Linoleic acid (LA) (99%), soybean lipoxygenase (soybean LOX type 1-B, EC 1.13.11.12), and xylenol range sodium salt were purchased from Sigma (St. Louis, MO, USA). All other reagents were of analytical grade and were commercially available.

### 3.2. Immobilization of Soybean LOX on Nanoporous Rice Husk Silica (RHS) by Adsorption

Nanoporous rice husk silica (RHS) was prepared as described previously [29,33], and was obtained from Nanobio Research Center (Jeollanamdo, South Korea). To immobilize soybean LOX on nanoporous RHS by adsorption, appropriate amounts of soybean LOX (0.2–12 mg) were mixed with 200 μL 50 mM sodium phosphate (pH 7.5) containing 20 mg RHS. After rocking at 100 cycles/min for 30 min at room temperature, RHS-adsorbed soybean LOX was separated from the supernatant by centrifugation at 12,000× *g* for 1 min at room temperature. Unadsorbed protein was quantified and the amount of soybean LOX adsorbed on RHS was calculated by subtracting the amount of protein in the supernatant from the total amount of loaded protein. Protein concentration was determined by the bicinchoninic acid (BCA) method [54] with bovine serum albumin (BSA) as the standard.

### 3.3. Assay for Soybean LOX Activity

The activity of free soybean LOX in solution at a high concentration of substrate was determined continuously by measuring the concentration of conjugated dienes in hydroperoxyl fatty acids as described previously [48]. Briefly, the assay was performed in a 2.5 mL solution containing 50 mM Tris-HCl (pH 7.2), 2.5 mM LA, 0.2% Tween 20 (*v/v*), and 0~0.2 mg soybean LOX. LOX activity was determined at 25 °C by monitoring the absorbance of HPOD (hydroperoxy octadecadiienoic acid) at 234 nm (ε = 25,000 M^−1^cm^−1^). The spectrophotometric continuous assay described above for free soybean LOX could not be employed to assay RHS-immobilized LOX because the insoluble RHS matrix exists as an emulsion. Therefore, discontinuous assay was performed using the ferrous oxidation–xylenol orange method [55]. Aliquots were taken from reaction mixtures (100 µl) containing RHS-immobilized soybean LOX at appropriate time intervals and mixed with the xylenol orange reagent (125 µM xylenol orange, 100 mM sorbitol, 25 mM ferrous ammonium sulfate, and 25 mM sulfuric acid) in a total volume of 1 mL. The assay mixture was incubated at room temperature for 45 min and centrifuged to remove the RHS matrix. The supernatant was collected and absorbance at 560 nm was measured. The concentration of HPOD was calculated using the extinction coefficient (ε = 267,000 M^−1^cm^−1^) for hydrogen peroxide at 560 nm.

### 3.4. Desorption Kinetics of RHS-Adsorbed Soybean LOX

To investigate the stability of the RHS-immobilized soybean LOX during storage, soybean LOX (~250 μg) adsorbed on RHS (12 mg) in 50 mM Tris-HCl (pH 7.2) or 50 mM sodium phosphate (pH 7.2) was stored at 4 °C and analyzed for 12 weeks. At weekly intervals, RHS-adsorbed soybean LOX was vortexed for 3 s and then centrifuged at 12,000× *g* for 1 min at room temperature, and precipitated RHS was resuspended in the same buffer. The amount of desorbed soybean LOX was estimated by measuring protein concentration by the BCA method.

### 3.5. Effect of pH on RHS Adsorption Efficiency of Soybean LOX

Soybean LOX (~150 μg) was adsorbed on RHS (2 mg) by mixing and rocking at 100 cycles/min for 30 min at room temperature in pH-specific buffers. The mixtures were centrifuged at 12,000× *g* for 1 min at room temperature. Soybean LOX in the supernatant was quantified using the BCA method. The following buffers were used: 50 mM citric acid (pH 5.06/5.55), 50 mM sodium phosphate (pH 6.07/6.54/7.04/7.51), 50 mM Tris-HCl (pH 8.03/8.57), and 50 mM boric acid (pH 9.01/9.45/10.01/10.51/11.04).

### 3.6. Recycling of RHS-Adsorbed Soybean LOX

RHS-adsorbed soybean LOX was recovered after each cycle by centrifugation at 12,000× *g* for 1 min at room temperature. LOX reactions were carried out with rocking at 100 cycles/min for 30 min at room temperature using a slight modification of the discontinuous assay described above. The recovered LOX (~30 μg per 2 mg RHS) was mixed with 200 µL 50 mM sodium phosphate (pH 7.2) and added to 1.2 mL 50 mM sodium phosphate (pH 7.2) containing 100 µM LA and 0.008% Tween 20 (*v/v*) at room temperature. Aliquots of the assay mixture (50 µL) were taken at 5, 10, 20, 40, 60 and 80 s, mixed with 1 mL xylenol orange reagent to stop the enzyme reaction, and were incubated at room temperature for 45 min. HPOD was quantified by absorbance at 560 nm. After the completion of each reaction cycle, the remaining solution was washed with 1 mL 50 mM sodium phosphate (pH 7.2) and recovered by centrifugation at 12,000× *g* for 1 min at room temperature. The recovered RHS-adsorbed LOX was reused and reassayed as described. Protein in the supernatant after each washing step was determined by the BCA method.

### 3.7. Temperature Dependence of RHS-Adsorbed Soybean LOX Activity

The temperature dependence of RHS-adsorbed soybean LOX catalysis was determined as follows: RHS-immobilized soybean LOX (~50 μg LOX/2 mg RHS) was suspended in 100 μL 50 mM sodium phosphate (pH 7.2), incubated in a water bath at 25, 30, 35, 40 and 45 °C for 30 min, chilled on ice, and then assayed for remaining activity. Enzyme activity was determined by the discontinuous assay method.

## 4. Conclusions

To develop a facile and practical method to immobilize enzymes for synthesis of oxylipins in vitro, soybean LOX was immobilized on nanoporous RHS with a maximal percentage of adsorption of 50%. The RHS-adsorbed soybean LOX retained 73% of the activity of the free form, which is an improvement over the performance of soybean LOX covalently immobilized on RHS. The adsorption efficiency of soybean LOX on RHS was optimal at pH 7.2. The operational stability allowed the RHS-adsorbed soybean LOX to be used reproducibly for at least five cycles, and the increased thermal stability permitted the durable use of the RHS-adsorbed soybean LOX at a moderately high temperature. However, covalent linking may still provide practical advantages for recycling RHS-immobilized enzymes. In summary, nanoporous RHS is an inexpensive and environment-friendly matrix for enzyme immobilization, and adsorption on RHS is a simple and rapid method with a demonstrated excellent outcome for soybean LOX/RHS. Immobilization on RHS by adsorption may provide a convenient and facile method for the immobilization of many other proteins of interest.

## Data Availability

The data presented in this study are openly available reference number [29,33,35].

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
