# Peer review of "Immobilization of Soybean Lipoxygenase on Nanoporous Rice Husk Silica by Adsorption: Retention of Enzyme Function and Catalytic Potential"

_molecules, 2021, doi:10.3390/molecules26020291_

Round 1
Reviewer 1 Report
The topic of the manuscript “Immobilization of Soybean Lipoxygenase on Nanoporous Rice Husk Silica by Adsorption: Retention of Enzyme Function and Catalytic Potential” sounds interesting. However, in the current form it is unsuitable for publication in Molecules. Before the manuscript might be considered for publication, I suggest a major revision.
In paragraph 2.1, the sentence Mesoporous materials are useful matrices for in vitro biosynthesis, at least in part because of their porous nature, which increases the amount of surface area available to interact with enzymes, leading to higher enzyme content per unit mass than alternative nonporous reaction matrices.” Needs some references. See for example Sannino, F.; Costantini, A.; Ruffo, F.; Aronne, A.; Venezia, V.; Califano, Nanomaterials 2020, 10, 108; Venezia, V.; Sannino, F.; Costantini, A.; Silvestri, B.; Cimino, S.; Califano, V. Micropor. Mesopor. Mat. 2020, 110203; Pang, J.; Zhou, G.; Liu, R.; Li, T. Mater. Sci. Eng. C 2016, 59, 35-42; Diaz, J. F.; Balkus Jr, K. J. J. Mol. Catal. B Enzym. 1996, 2, 115-126. Washmon-Kriel, L.; Jimenez, V. L.; Balkus Jr, K. J. J. Mol. Catal. B Enzym. 2000, 10, 453-469.
“RHS contains nanopores approximately 4–5 nm diameter”. Pores of 4-5 nm are commonly referred as mesopores.
When an enzyme is physically immobilized on a porous support, it is important that the pore size match the molecular diameter of the enzyme. Hence, it would be appropriate to indicate the molecular size of the immobilized enzyme.
The authors should better justify why the unadsorbed enzyme should have a different behaviour respect to the free enzyme.
Pag 6 “Catalytic turnover (kcat) decreased to 11% of the kcat of free soybean LOX, as expected”. Why was it expected? How this drastic decrease of Kcat is explained?
Dissociation kinetics is ambiguous. It should be indicated as leaching or desorption.
Pag. 5: “free in solution in the presence of RHS with mixing (unadsorbed LOX)” I don’t understand the significance of this experiment, since the free enzyme is solution should be adsorbed during this run, considering that the adsorption kinetic is rapid.
Page 6 paragraph 2.3: “The amount of unadsorbed LOX was 70.12 g (27.61%) in Tris-HCl (pH 7.2) and 16.88 g (6.78%) in 50 mM sodium phosphate (pH 7.2), such that 72.39% or 93.22% of the enzyme input was adsorbed onto RHS”. This is in contrast with the results reported in fig. 1, which indicate that the maximum adsorption efficiency is 50%.
Since most analysis are carried out at pH 7.2, why in figure 3 the experimental point at pH 7.2 is missing?
Experimental section: why the enzyme/support ratio varies depending on the parameter to be determined (dissociation kinetic 0.02; pH stability 0.075; operational stability 0.015; thermal stability 0.025)? In conclusion, the authors did not propose an optimized biocatalyst.
Reviewer 2 Report
The authors show a biocatalytical in vitro characterization of an immobilized enzyme on RHS matrix. Previously the authors already showed a covalent immobilization which had the great disadvantage of loosing a lot ov activity of the enzyme. Therefore here, the unspecific adsorpion of enzyme on RHS matrix was investigated.
Overall I would like to see more accuracy in the performance of experiments also in terms of conclusions based on data without error bars, so no indication of how accurate the values on which basis a conclusion could be drawn. Therefore I suggest strongly to work on the alignment of experimental statements and experimental data including the addition of error margins and significance analysis.
Intro:
Minor:
- 'Most of these studies report unacceptable loss of enzyme stability
and function' - phase softer
Results:
Minor
- 2.3
Please fix all digits of all values after the comma here. I cannot fully judge the meaningful digits of this experiments as no error margins are given but I suspect one can greatfully round these values to 70µg and 30% in Tris and 17µg and 7% in sodium phosphate buffer.
Please provide error bars for your data.
Overall: could you make the numbers in the figures a bit bigger so that they are better readable?
- 2.5
In your interpretation of the data you referred to relative % of activity and compare it to the covalently bound activity. Can you Measure them in parallel and also show absolute values to clarify which form is the better performer.
Major:
Results:
- 2.1:
Figure 1B: Maximal adsorption capacity looks like 26% for LOX as graph is reaching 13% at 0.5 ratio on x-axis, extrapolated to 1.0 this means about 26%. Could you double check
Also Material methods plus figure legend suggest, that 14 mg protein were used on 20 mg RHS, which is a ratio of 0.7 – a value that is not present in graph 1B. Could you explain?
I do not see why a similar isoelectric point is leading to the conclusion that the adsorption of the proteins involves electrostatic and hydrophobic interactions. Until shown different this is true for all proteins and materials. Consider to take this argumen out. It is not proven by your experiment. Also this is not the main message of this section: it is: both bind and some protein gets adsorbed
- Figure 1:
How well can you measure this amounts of proteins, please include error bars
Make y-axis equal for A and B for better comparison
Why were not conditions chosen with protein access over RHS? As this could saturate the solid matrix much more.
- 2.3
Please fix all digits of all values after the comma here. I cannot fully judge the meaningful digits of this experiments as no error margins are given but I suspect one can greatfully round these values to 70µg and 30% in Tris and 17µg and 7% in sodium phosphate buffer.
Why are there two paragraphs, it is not clear to me, what the difference is. Can you merge them and double check the numbers?
Please provide error bars for your data to show that this is a significant difference.
Without error bars it is hard to judge if the one performs better than the other. So please soften your interpretation here. I think they perform equal. If you would compare 5 different sets of immobilized enzyme next to each other in these two buffer systems.
Clarify numbers on the Y-axis: This is the total LOX amount (so the sum of all LOX measured) each week, right? The plateau reflects therefore the total amount of protein that was bound (initially 250µg but only how much would be bound at this protein/RHS ratio?)? Or do you think the rest of the protein is still bound and dissociation and re-binding comes to an equilibrium here?
Do not call it a thermodynamic/kinetic stability as you are not measuring any thermodynamic properties of the system.
- 2.4
I am missing the point where the data from 2.3 is consistent with the data from 2.3? Could you elaborate. Do you mean Figure 1?
Can you critically reflect on the fact that you also might precipitate your protein at harsh pH values
- 2.5
If free enzyme 40degC is an outlier, the enzymes essentially behave identical. Could you repeat experiments and add error bars then the significance might appear. Right now I would say it is hard to say which data points to believe in.
Discussion:
Minor:
Is the dissociation of the enzyme from the matrix an issue for the follow up process of the products as they appear as contaminants?
Can the matrix be reused for technical application (reload with enzyme)
Please check digits after the comma as they are not all relevant.
Major:
Double check all conclusions: e.g. I don’t think the tested buffers make a big difference considering the error bars for the experiment. So the statement that anionic buffer is better than cationic buffer is not underlined by the experiment.
Please add a discussion if this activity and reusability is sufficient for industrial applications. A reloading every 5 cycles might not be suitable for such workflows and therefore the adsorption might be better than covalent linkage, but overall not of practical relevance.
Round 2
Reviewer 1 Report
In the final version of the manuscript, the authors addressed all the suggestions made. It can be published in the present form.
Reviewer 2 Report
All my raised concerns where addressed by the authors.